# Vox-Infinity: Benchmarking the Limits of Long-Context Spoken Language Models

## Abstract

Long-context reasoning remains a fundamental challenge for large language models, as excessively long inputs often lead to the forgetting of salient information. This issue is even more pronounced in the speech domain, where audio, as a low-compression modality, requires significantly more embeddings than text to preserve both semantic content and acoustic cues. To address this, we introduce **Vox-Infinity**, the first benchmark specifically designed to evaluate long-context understanding in spoken language models. Vox-Infinity systematically extends audio history along two dimensions: turns and duration. It covers a diverse range of representative scenarios, including dialogues with varying structural depth and semantic complexity. Crucially, it provides explicit answer provenance annotations and organizes samples based on the context length required to resolve each query, enabling precise and length-aware evaluation of model performance. Furthermore, we present the first comprehensive study of history modeling strategies in this setting, analyzing how models balance the trade-off between preserving long-range semantics and retaining recent acoustic signals. Cases and datasets are available at `https://vox-infinity.github.io`.

## 1 Introduction

Spoken language systems (Kaplan & Haenlein, 2019; Hachman, 2019; Chu et al., 2023; 2024; Ghosh et al., 2025; Chen et al., 2024a) aim to engage in intelligent and natural interactions with humans, requiring the ability to comprehend not only semantic content but also paralinguistic cues embedded in speech. Early systems (Kaplan & Haenlein, 2019; Hachman, 2019) primarily relied on automatic speech recognition (ASR) (Yu & Deng, 2016) to convert audio into text, and achieved basic dialogue functionality through intent recognition (Liu et al., 2019; Zheng et al., 2017) and dialogue state tracking (Williams et al., 2016; Jacqmin et al., 2022) based on the transcribed content. With the emergence of large language models (LLM) (Bai et al., 2023a; Dubey et al., 2024; Fang et al., 2025; Yang et al., 2025), some systems (SpeechTeam, 2024; Chen et al., 2025; Zhang et al., 2024a) have significantly improved in semantic understanding. Some recent approaches have integrated ASR and LLM to construct powerful multi-stage dialogue systems with strong interactive capabilities. However, as cascaded architectures, such systems (Zhang et al., 2024a; Chen et al., 2025) still struggle to fully capture and interpret the rich information present in speech, particularly acoustic features such as intonation, emotion, and emphasis. To overcome these limitations and advance audio understanding, recent studies (Tang et al., 2023; Xu et al., 2025; Chen et al., 2024a; Ghosh et al., 2025; Zhang et al., 2024b) have proposed large audio language models that incorporate raw audio directly into the dialogue framework. This end-to-end integration allows models to better recognize and utilize nuanced acoustic signals (Ao et al., 2024; Cheng et al., 2025) that are often lost in ASR-based pipelines.

However, audio presents a significant challenge due to its low information density (Ji et al., 2024a). Spoken content typically requires 12 to 25 embeddings (Zeng et al., 2024; Ji et al., 2024b) per second to encode, resulting in much longer input sequences than text for conveying equivalent semantic information. This substantially increases the effective context length in spoken language tasks and exacerbates the difficulty of long-context modeling. Prior researches (Bai et al., 2023b; Ding et al., 2025) on long-context language models has shown that performance tends to degrade as relevant information moves further back in the input—a limitation that similarly impacts spoken dialogue systems. While long-context reasoning has been widely explored in textual (Ding et al., 2024) and

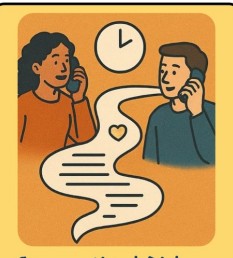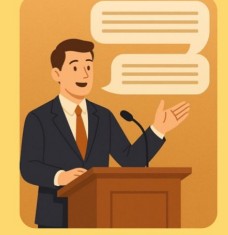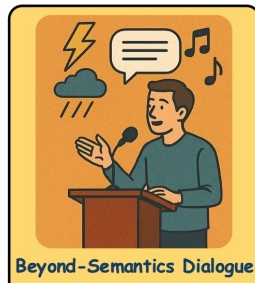

**Ultra Multi-turn Dialogue**
- Customer Service
- Tech Support
- Requirement Gathering

**Conversational Dialogue**
- Phone Chatting
- WeChat Long Conversation

**Personal Monologue**
- Public Speech
- Lecture
- Storytelling

**Beyond-Semantics Dialogue**
- Game Interaction
- Movie Dialogue with Sound Effects

Figure 1: Illustration of four representative dialogue scenarios in VOX-INFINITY. **(a) Ultra Multi-turn Dialogue**: frequent back-and-forth exchanges for verifying requirements and details. **(b) Conversational Dialogue**: extended, casual conversations emphasizing sustained interaction and advice. **(c) Personal Monologue**: single-speaker, long-form speech such as lectures or presentations. **(d) Beyond-Semantics Dialogue**: conversations enriched with acoustic information (e.g., environmental sounds, sound effects, music) beyond pure semantics.

visual modalities (Chen et al., 2024b; Zhang et al., 2024b), research in the spoken language domain remains in its early stages. Most existing efforts (Goel et al., 2025) focus on extending audio input length beyond 30 seconds, a constraint largely imposed by the capacity of audio encoders such as Whisper (Rouditchenko et al., 2024). However, a systematic evaluation of spoken language models under varying long-context conditions remains notably lacking.

To bridge this gap, we introduce VOX-INFINITY, the first benchmark specifically designed to evaluate long-context understanding in spoken language models under various context lengths. Vox-Infinity systematically extends audio history along two dimensions: the number of dialogue turns and the duration of each turn. As illustrated in Figure 1, the benchmark encompasses a diverse range of realistic scenarios, including ultra multi-turn dialogues, conversational dialogues, personal monologues, and beyond-semantic dialogues. To enable precise evaluation, each question is annotated with the exact location of its corresponding answer within the history audios. This allows us to measure the minimum context length required for accurate resolution. Recognizing that model performance may vary significantly depending on the length of relevant context (Bai et al., 2024), Vox-Infinity groups test samples based on the position of the answer within the dialogue context, enabling fine-grained performance analysis across different long-context lengths. In addition, spoken language models can adopt various strategies for modeling historical information, such as text-only (Chen et al., 2024a; Wu et al., 2025), audio-only (Zeng et al., 2024), or hybrid modality approaches. We conduct the first comprehensive study of these strategies under long-context conditions. Our findings highlight how models navigate the trade-off between retaining long-range semantic content and preserving recent acoustic cues, offering new insights into the development of future spoken dialogue systems capable of efficient and effective long-context reasoning. The Vox-Infinity will be released at the demo page `https://vox-infinity.github.io`. Our main contributions are as follows:

- We introduce Vox-Infinity, the first benchmark specifically designed to evaluate long-context reasoning under varying context conditions in spoken language models. It systematically scales dialogue history along two axes (i.e., turn count and turn duration) and covers a diverse range of realistic long-context spoken language understanding scenarios.

- Vox-Infinity provides explicit answer provenance annotations and groups test samples based on the minimum context length required for resolution. This enables precise, length-sensitive evaluation of model performance across varying long-context regimes, which is critical for diagnosing model limitations and benchmarking progress.

- We conduct the first in-depth analysis of history modeling approaches for long-context spoken language understanding. By comparing text-only, audio-only, and hybrid-modality strategies, we reveal how models navigate the trade-off between retaining long-range semantic information and preserving recent acoustic cues, offering insights for future spoken dialogue system design.

## 2 RELATED WORKS

### 2.1 SPOKEN DIALOGUE SYSTEMS

The rapid advancement of large language models (LLMs) has fueled the emergence of increasingly powerful spoken dialogue systems. A notable early milestone was SpeechGPT (Zhang et al., 2023), the first speech-centric LLM to integrate discrete speech units into a unified language modeling framework. Building on this foundation, large-scale audio language models such as Qwen-Audio 1/2 (Chu et al., 2023; 2024) extended capabilities to over 30 audio-related tasks, including speech recognition, speech translation, and audio event detection—laying the groundwork for more sophisticated spoken language understanding. These core audio modeling capabilities have since enabled the development of specialized dialogue systems. For example, StyleTalk (Lin et al., 2024) focuses on emotional dialogue, introducing the first model capable of generating speech with distinct emotional prosody. More recently, models like GLM-4-Voice (Zeng et al., 2024), LLAMA-Omni 1/2 (Fang et al., 2024; 2025), and Qwen2.5-Omni (Xu et al., 2025) adopted discrete speech units as output tokens, enabling fully end-to-end spoken dialogue systems without relying on intermediate text representations.

Despite these advancements, most current systems remain constrained by limited context windows. Because audio is a low-compression modality, representing equivalent semantic content requires much longer sequences than text, posing serious challenges for long-context modeling. To address this, Audio-Flamingo 2/3 (Ghosh et al., 2025; Goel et al., 2025) pioneered long-context modeling for spoken dialogue, extending context length beyond 30 seconds. In parallel, models such as Slam-Omni (Chen et al., 2024a) and Step-Audio2 (Wu et al., 2025) alleviated audio compression bottlenecks by incorporating text-based history representations, which significantly improve memory efficiency and dialogue history retention.

In this study, we build upon prior work to conduct the first systematic comparison of spoken language models under varying long-context conditions, aiming to advance the development of long-context spoken language understanding. Furthermore, we perform an in-depth evaluation of different history modeling strategies to identify the most effective method for context representation. Our goal is to strike a balance between retaining long-range semantic information and capturing recent acoustic cues, both of which are essential for robust spoken dialogue understanding.

### 2.2 LONG-CONTEXT MODELING AND BENCHMARKS

Long-context modeling (Li et al., 2024) has emerged as a central research area in the development of LLMs, as the ability to process extended inputs is essential for maintaining discourse coherence, recalling earlier information, and performing complex reasoning. A broad range of techniques have been proposed to improve long-context capabilities, including architectural modifications and advanced positional encoding schemes. For instance, models such as Kimi (Team et al., 2025) leverage rotary position embeddings (Su et al., 2024) and attention optimization to scale up to contexts of tens of thousands of tokens. Some have explored position editing (He et al., 2024) and information compression (Jiang et al., 2023), dynamically updating the retained context to reduce memory load while preserving relevant content.

To systematically evaluate these capabilities, a number of long-context benchmarks have been developed. Early efforts like LAMBADA (Paperno et al., 2016) and Long Range Arena (Tay et al., 2020) focused on synthetic or controlled settings that test models' ability to handle long-range dependencies. More comprehensive benchmarks such as LongBench (Bai et al., 2023b; 2024) and L-Eval (An et al., 2023) have expanded the task set to include summarization, question answering, logical reasoning, and dialogue. These studies consistently show that even state-of-the-art LLMs struggle to maintain fidelity and relevance when the context window exceeds 16k or 32k tokens.

Despite these advancements, existing evaluations remain almost exclusively text-only, overlooking the distinctive challenges of spoken language, such as acoustic cues (Ao et al., 2024), and significantly longer sequence lengths due to audio's low information density (Ji et al., 2024a). These factors make spoken language understanding a fundamentally different long-context problem. To address this gap, we propose Vox-Infinity, the first benchmark explicitly designed to evaluate long-context reasoning in spoken language models. Vox-Infinity scales audio history along two axes (turn count and duration) and incorporates a diverse set of realistic spoken scenarios. By providing de-

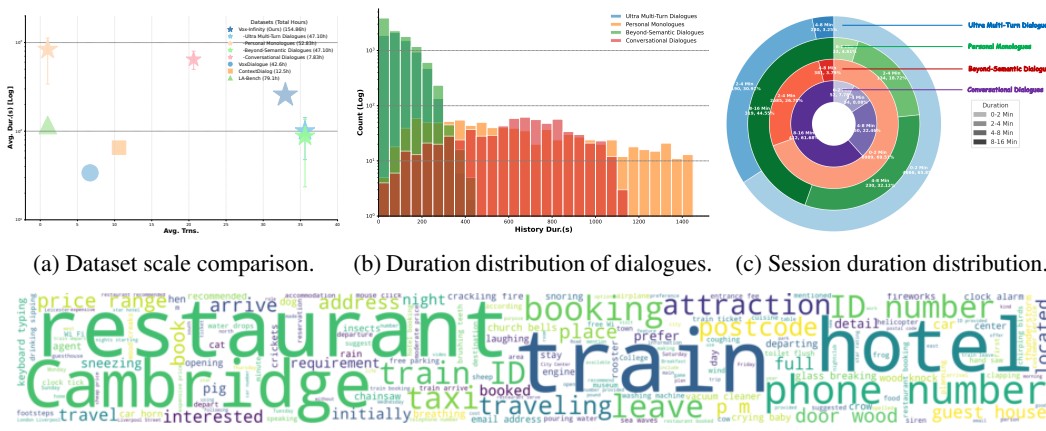

(a) Dataset scale comparison.  (b) Duration distribution of dialogues.  (c) Session duration distribution.

(d) Word cloud visualization of common topics.

Figure 2: Statistics and characteristics of VOX-INFINITY.

tailed annotations and organizing test cases by answer provenance and context length, Vox-Infinity enables fine-grained context-aware evaluation, pushing the boundaries of long-context modeling beyond the text domain to better reflect real-world usage of spoken language systems.

# 3 VOX-INFINITY: LONG-CONTEXT SPOKEN LANGUAGE BENCHMARK

## 3.1 DATASET CONSTRUCTION

**Stage 1: Long-Context Spoken Dialogue Generation.** VOX-INFINITY encompasses four distinct spoken dialogue scenarios. *(I) Ultra Multi-Turn Dialogues* consist of conversations with more than 20 turns, constructed from SpokenWOZ (Si et al., 2024), a task-oriented dialogue dataset recorded by real speakers. *(II) Conversational Dialogues* are multi-turn conversations in which most turns exceed 30 seconds; we generate such long dialogues with GPT on diverse topics and concatenate them to simulate daily phone calls and casual discussions. We used CosyVoice2-0.5B (Du et al., 2024) for speech synthesis. To ensure consistent speaker timbre, we followed prior work (Cheng et al., 2025) and applied a speaker verification system (Plaquet & Bredin, 2023). All synthesized audio was further validated using Whisper (Rouditchenko et al., 2024) to guarantee correct pronunciation. *(III) Personal Monologues* feature single-speaker speeches longer than two minutes, derived from GigaSpeech (Chen et al., 2021). Based on timestamp annotations, we reconstruct short segments into their original long-form monologues, forming a benchmark for long-duration speech understanding. *(IV) Beyond-Semantic Dialogues* extend beyond purely textual content by incorporating concrete acoustic events. Specifically, we integrate audio from the ESC-50 dataset (Piczak, 2015) into multi-turn dialogues, enriching the benchmark with a beyond-semantic information.

**Stage 2: Question–Answer Pair Generation.** In the second stage, we generate multiple question–answer (QA) pairs for each dialogue to evaluate model understanding under varying context lengths. Since model responses may evolve as the dialogue progresses, we employ GPT-4 to generate diverse questions based on the whole dialogue to ensure each yields a unique and unambiguous answer. To precisely control the required context, GPT-4 is also instructed to specify which dialogue turns must be referenced when answering each question. Detailed prompt design and generation strategies are provided in the appendix B.1.

**Stage 3: Data Verification and Quality Control.** Finally, to guarantee dataset quality, each QA pair undergoes human verification to confirm that the answer is both correct and unique within the given dialogue context. Any question found to have ambiguous or incorrect answers is regenerated and revalidated until it passes manual validation. Through this rigorous process, we construct a high-quality long-context spoken dialogue dataset covering diverse scenarios and tasks.

Table 1: Comparison of existing long-context benchmarks across textual and audio modalities. **Avg. Tok.** denotes the average number of tokens per session, **Avg.Dur.** denotes the average duration per session, and **Avg.Trns.** denotes the average number of turns per session. **Prov.** indicates whether answer provenance annotations are available. **Length distribution** refers to the context-length ranges used for evaluation; ‡ for datasets without explicit definitions, we adopt the near-upper quartile of context length as the upper bound. † For audio benchmarks, token counts are estimated using a normal count of 25 tokens per second.

| Benchmarks | Session | QAs | Dur.(h) | Avg. Tok.† | Avg. Dur.(s) | Avg. Trns. | Prov. | Length Distribution‡ |
|---|---|---|---|---|---|---|---|---|
| *Long Context Benchmark (Textual)* | | | | | | | | |
| LongBench (Bai et al., 2023b) | 4,550 | 4,550 | - | 8.1k | - | 1.00 | ✗ | 0-4k, 4k-8k, 8k+ |
| ∞ BENCH (Zhang et al., 2024c) | 3,946 | 3,946 | - | 200k | - | 1.00 | ✗ | 0-200k |
| LongBench-V2 (Bai et al., 2024) | 503 | 503 | - | 54k | - | 1.00 | ✗ | 0-32k, 32k-128k, 128k+ |
| *Long Context Benchmark (Audio)* | | | | | | | | |
| VoxDialogue (Cheng et al., 2025) | 4,526 | 4,526 | 42.56 | 0.8k | 33.9 | 6.70 | ✗ | 0-2min |
| ContextDialog (Kim et al., 2025) | 653 | 2,612 | 12.5 | 1.1k | 65.2 | 10.60 | ✗ | 0-2min |
| LA-Bench (Kong et al., 2024) | 2,429 | 2,429 | 79.12 | 2.9k | 117.3 | 1.00 | ✗ | 0-3min+ |
| BLAB (Ahia et al., 2025) | 1,176 | 1,600 | 833 | 77.2k | 3,088.8 | 1.00 | ✗ | 0-60min+ |
| Vox-Infinity (ours) | 2,174 | 19,032 | 154.86 | 6.4k | 256.4 | 32.98 | ✓ | 0-2min, 2-4min, 4-8min, 8min+ |
| - *Ultra Multi-Turn Dialogue* | 991 | 7,086 | 47.10 | 4.3k | 171.1 | 35.89 | ✓ | 0-2min, 2-4min, 4-8min |
| - *Beyond-Semantic Dialogue* | 991 | 10,055 | 47.10 | 4.3k | 171.1 | 35.89 | ✓ | 0-2min, 2-4min, 4-8min |
| - *Conversational Dialogue* | 33 | 681 | 7.83 | 21.4k | 854.2 | 12.21 | ✓ | 0-2min, 2-4min, 4-8min, 8min+ |
| - *Personal Monologues* | 159 | 1,210 | 52.83 | 30.0k | 1,196.2 | 1.00 | ✓ | 0-2min, 2-4min, 4-8min, 8min+ |

## 3.2 DATASET STATISTICS

**Detailed Statistics of Vox-Infinity.** To provide a comprehensive overview of Vox-Infinity, Figure 2 presents detailed dataset statistics. As shown in Figure 4a, we compare the four subsets of Vox-Infinity with existing long-audio benchmarks in terms of average number of turns and total dialogue duration. The results demonstrate that Vox-Infinity exhibits substantially longer conversations both in turns and duration, better reflecting the challenges of real-world long dialogues. Figure 4b illustrates the distribution of dialogue history lengths across the four subsets. Vox-Infinity spans a broad temporal range, from short to long conversations, making it suitable for evaluating systems under different context-length conditions. Figure 4c further shows the proportion of samples across different length ranges within each subset, highlighting the benchmark's diversity in temporal coverage. Finally, Figure 4d presents a word cloud of common topics in Vox-Infinity. The dataset features a high concentration of queries targeting key contextual information—such as phone numbers, train IDs, and restaurants—all of which have explicit answers, making them ideal for assessing a model's ability to retain and utilize long-term contextual memory. In addition, the dataset includes queries about acoustic events, such as clock alarms and keyboard typing, which emphasize the distinctive challenges of spoken dialogue compared to purely textual dialogue.

**Comparison with Existing Benchmarks.** Table 1 situates Vox-Infinity within the landscape of long-context benchmarks across text and audio modalities. Text-based benchmarks such as Long-Bench (Bai et al., 2023b; 2024) enable performance evaluation under varying context lengths. In contrast, existing audio benchmarks (Goel et al., 2025) primarily aim to overcome the 30-second input limit of encoders like Whisper (Radford et al., 2023), focusing on extending input length rather than analyzing performance across different context regimes. Vox-Infinity fills this gap by being the first audio benchmark to explicitly evaluate models under multiple context-length conditions, allowing fine-grained analysis of long-context reasoning. Compared with benchmarks like BLAB (Ahia et al., 2025), which focus on hour-long audio, Vox-Infinity adopts more practical dialogue durations while preserving contextual diversity. This makes it better suited for realistic and scalable spoken dialogue evaluation.

Another limitation of prior QA-based benchmarks (Bai et al., 2024; Goel et al., 2025) is the absence of answer provenance annotations, which makes the true amount of context required to answer a question ambiguous. Vox-Infinity addresses this issue by explicitly labeling provenance, enabling precise measurement of the actual context length needed to support each answer. By jointly consid-

Table 2: Performance comparison of different spoken language models across various dialogue scenarios and context lengths. The *Conversational* and *Beyond-Semantic* are in choice tasks, while the *Ultra Multi-Turn* and *Personal Monologues* are in an open-ended setting. *m* stands for minutes. *avg* denotes the average accuracy of the corresponding subset across various context-length distributions. All values are reported as percentages. Models marked with † are closed-source.

| Method | Ultra Multi-Turn | | | | Conversational | | | | | Personal Monologues | | | | | Beyond-Semantic | | | |
|---|---|---|---|---|---|---|---|---|---|---|---|---|---|---|---|---|---|---|
| | 0-2m | 2-4m | 4-8m | avg | 0-2m | 2-4m | 4-8m | 8m+ | avg | 0-2m | 2-4m | 4-8m | 8m+ | avg | 0-2m | 2-4m | 4-8m | avg |
| *Text-as-history modeling strategy.* | | | | | | | | | | | | | | | | | | |
| Qwen2-Audio | 35.4 | 33.9 | 31.5 | 33.6 | 41.2 | 38.8 | 34.1 | 32.4 | 36.6 | 33.5 | 26.2 | 23.6 | 18.1 | 25.4 | 7.2 | 6.7 | 7.4 | 7.1 |
| GLM-4-Voice | 45.8 | 42.2 | 41.2 | 43.1 | 44.2 | 48.2 | 43.3 | 29.4 | 41.3 | 29.0 | 22.6 | 26.9 | 14.3 | 23.2 | 7.1 | 6.2 | 7.9 | 7.2 |
| Step-Audio2 | 52.1 | 45.3 | 41.9 | 46.4 | 80.5 | 77.8 | 74.1 | 75.2 | 76.9 | 84.2 | 78.1 | 73.6 | 71.4 | 76.8 | 7.3 | 6.9 | 7.1 | 7.1 |
| Mimo-Audio | 65.0 | 66.4 | **67.7** | 66.4 | 82.7 | **92.1** | 82.1 | **79.3** | 81.0 | 89.7 | 85.2 | **84.2** | 72.6 | 82.9 | 6.8 | 6.9 | 7.2 | 7.0 |
| Qwen2.5-Omni | **69.3** | **68.2** | 66.4 | 68.0 | **84.6** | 90.7 | **83.3** | 77.1 | 83.9 | **92.1** | **91.7** | 79.0 | **78.3** | 85.3 | 6.7 | 7.1 | 7.0 | 6.9 |
| †GPT-Realtime | 63.2 | 61.9 | 59.4 | 61.5 | 81.7 | 82.1 | 71.9 | 65.4 | 75.3 | 73.1 | 69.2 | 66.9 | 65.2 | 68.6 | 6.7 | 7.2 | 6.9 | 6.9 |
| *Audio-as-history modeling strategy.* | | | | | | | | | | | | | | | | | | |
| Qwen2-Audio | 28.7 | 21.3 | 14.4 | 21.5 | 21.0 | 18.5 | 19.7 | 6.9 | 16.5 | 24.1 | 15.2 | 8.9 | 3.1 | 12.8 | 13.9 | 12.5 | 15.2 | 13.9 |
| GLM-4-Voice | 38.1 | 24.0 | 6.4 | 22.8 | 17.3 | 16.7 | 21.3 | 5.8 | 15.3 | 19.2 | 12.9 | 7.7 | 0.0 | 10.0 | 9.2 | 8.3 | 7.9 | 8.5 |
| Step-Audio2 | 45.8 | 37.9 | 32.4 | 38.7 | 74.2 | 71.5 | 68.3 | 64.9 | 69.7 | 73.9 | 38.2 | 17.6 | 13.4 | 35.8 | 15.4 | 13.1 | 12.5 | 13.7 |
| MiMo-Audio | 35.7 | 30.6 | 25.9 | 30.7 | **90.4** | **94.4** | **85.3** | **86.6** | 89.2 | 78.1 | 41.7 | 15.8 | 9.8 | 36.4 | 22.1 | 20.4 | 21.7 | 21.4 |
| Qwen2.5-Omni | **60.2** | **58.9** | **57.4** | 58.8 | 88.5 | 77.8 | **85.3** | 79.6 | 82.8 | **83.2** | **50.0** | **52.6** | **49.0** | 58.7 | 17.3 | 15.9 | 15.7 | 16.3 |
| †GPT-Realtime | 55.3 | 52.4 | 49.6 | 52.4 | 84.2 | 79.8 | 73.5 | 68.1 | 76.4 | 76.2 | 43.6 | 21.3 | 11.8 | 38.2 | **31.4** | **25.1** | **22.9** | 26.5 |

ering context-length granularity and answer provenance, Vox-Infinity establishes a benchmark that is more analytically rigorous for long-context modeling.

## 3.3 EVALUATION SUITE

Following the evaluation suite of LongAudioBench (Ghosh et al., 2025), we adopt Accuracy as the primary evaluation metric, which measures the proportion of correctly answered samples among all test cases. Given the high diversity of responses generated by Spoken Dialogue Systems, we follow prior work (Duan et al., 2024; Ghosh et al., 2025) and employ GPT-4o (OpenAI, 2024) to preprocess the model outputs. Responses are categorized into three possible labels: Yes, No, and Unknown. The inclusion of the Unknown label allows GPT-4o to explicitly indicate uncertainty, offering insights into how often the model refrains from committing to a definitive answer. This design helps mitigate potential biases that could arise if GPT-4o were forced to choose Yes or No in ambiguous situations. Detailed prompt templates for this evaluation process are provided in Appendix B.2. In addition to accuracy, we also measure latency, which is a critical factor in spoken language models. Specifically, we calculate the time elapsed from obtaining the audio embedding to generating the first audio token.

## 4 BENCHMARKING LONG-CONTEXT SPOKEN LANGUAGE MODELS

### 4.1 BENCHMARKING SLMs ACROSS CONTEXT LENGTHS

#### 4.1.1 COMPARISON MODELS.

To systematically assess the performance of state-of-the-art spoken language models under varying context lengths, we conducted a benchmark evaluation of three recent representative systems: GLM4-Voice (Zeng et al., 2024), Qwen2.5-Omni (Xu et al., 2025), Mimo-Audio (Xiaomi, 2025), Qwen2-Audio (Chu et al., 2024), Step-Audio2 (Wu et al., 2025) and GPT-Realtime. Gemini-2.5-Pro (Comanici et al., 2025) was excluded from our study due to its inability to incorporate dialogue history. For each model, we measured performance using dialogue histories represented entirely as text or entirely as audio. Textual histories were derived via automatic speech recognition (ASR) using the open-source SenseVoice model (SpeechTeam, 2024). To ensure fairness and comparability, all models were evaluated with a standardized maximum context window of 32,768 tokens during inference.

#### 4.1.2 MAIN RESULTS

Table 2 presents a comparison of model performance across different subsets of Vox-Infinity. As context length increases, both text- and audio-based histories exhibit performance degradation, albeit

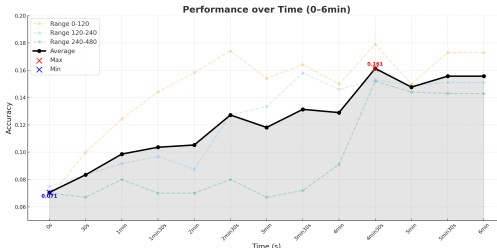

(a) Performance curves (accuracy) of different modeling strategies across on ultra multi-turn dialogue.

(b) Acoustic performance curves across different time intervals with varying audio input lengths.

Figure 3: Performance evaluation with Qwen2.5-Omni across different time intervals.

at different rates. For example, GLM-4-Voice with text history drops only moderately from 45.8% (0–2 min) to 41.2% (4–8 min) on Ultra Multi-turn Dialogue, whereas its audio-history counterpart declines much more steeply from 38.1% (0–2 min) to 6.4% (4–8 min). These results suggest that text-based models are more robust under long-context scenarios, while audio-based models are more sensitive to input length. However, for models with inherently stronger long-context capabilities (e.g., Qwen2.5-Omni and Mimo-Audio, whose base llms (Yang et al., 2025; Xiaomi et al., 2025) already supports extended contexts), such ability also transfers to the audio modality: when the sequence grows longer, their performance does not show obvious degradation. Figure 3a further illustrates the temporal trend of audio- vs. text-context modeling strategies: with shorter histories, the performance gap between text and audio is small, but as the dialogue progresses, naive audio-as-history modeling fails earlier, whereas enhanced audio-as-history strategies can sustain dialogue performance over longer spans.

Nonetheless, text-only modeling introduces inherent limitations. While text captures semantic content, audio additionally encodes prosodic and acoustic information; discarding the speech modality inevitably results in the loss of these cues. This limitation is particularly evident in the Beyond-Semantic Dialogue subset, where Qwen2.5-Omni achieves only 7.0 with text history but 15.7 with audio history, underscoring the critical role of acoustic signals. To further probe this effect, Figure 3b examines the impact of retaining different lengths of recent audio input. As shown, only models with access to audio can effectively leverage prosodic cues, yielding substantial improvements (e.g., Mimo-Audio reaches 22.1% on Beyond-Semantic with audio, compared to 6.8% with text). However, the gains plateau beyond approximately 4.5 minutes of audio, suggesting that current models remain challenged in exploiting very long acoustic contexts.

Finally, different models exhibit distinct strengths across task types. Ultra Multi-turn Dialogue and Personal Monologue are open-ended tasks, whereas Conversational Dialogue and Beyond-Semantic Dialogue are formulated as choice questions. These two formats emphasize different abilities: open-ended tasks require models to integrate and generalize information from the overall context, while choice tasks mainly demand contextual grounding and retrieval, which are comparatively simpler. As shown, performance on open-ended tasks declines more sharply with increasing context length, making them more challenging. In contrast, choice tasks show less pronounced degradation over time, yet clearly reveal the boundaries of contextual capacity—for instance, once the context exceeds 8 minutes, GLM-4-Voice's accuracy drops precipitously on the simple choice tasks. Moreover, the relative strengths of different models also diverge across these tasks: Qwen2.5-Omni demonstrates consistently strong performance on both open-ended and choice settings, whereas Mimo-Audio shows competitive contextual ability primarily in the choice tasks.

## 4.2 BENCHMARKING SLMs WITH DIFFERENT HISTORY MODELING STRATEGIES

### 4.2.1 COMPARISON HISTORY MODELING STRATEGIES.

**Mono-Modality History Modeling.** Spoken dialogue history modeling methods can be broadly categorized into audio-only and text-only approaches. For text-only models, we employ the pre-trained SenseVoice model to obtain transcriptions, which are then used as contextual information.

Table 3: Comparison of history modeling strategies in spoken dialogue systems. The table reports accuracy (ACC) across different context lengths (0–2 minutes, 2–4 minutes, and 4–8 minutes) for the *Ultra Multi-Turn Dialogues* together with average latency. History turns are denoted as $T$ (text) and $A$ (audio).

| ID | Method | Audio Turns Pre-QAs | Ultra Multi-Turn Dialogue | | | Latency (ms) ↓ |
|----|--------|---------------------|---------|--------|--------|----------------|
| | | | *0-2min* | *2-4min* | *4-8min* | |
| *Mono-Modality History Modeling* | | | | | | |
| E1 | Audio-only | $nA$ | 38.1% | 24.0% | 6.4% | 54 |
| E2 | Text-only | $nT$ | 45.8% | 42.2% | 41.2% | 45 |
| *Hybrid-Modality History Modeling* | | | | | | |
| E3 | Substitution | $(n-1)T + 1A$ | 27.8% | 23.6% | 10.6% | 110 |
| E4 | Re-computation | $(n-1)T + 1A$ | **46.4%** | 40.3% | **40.9%** | **75** |
| E5 | Mope (ours) | $(n-1)T + 1A$ | 45.8% | **41.4%** | 40.0% | 111 |
| E6 | Re-computation | $(n-3)T + 3A$ | 43.9% | 40.2% | 39.1% | 151 |
| E7 | Mope (ours) | $(n-3)T + 3A$ | **44.8%** | **40.6%** | **40.0%** | **122** |
| E8 | Re-computation | $(n-5)T + 5A$ | **44.1%** | 38.4% | 36.2% | 194 |
| E9 | Mope (ours) | $(n-5)T + 5A$ | 43.1% | **39.0%** | **38.4%** | **126** |

To ensure fairness, all experiments in this section are conducted with GLM-4-Voice, which supports both audio and text as historical context. More details are given in Appendix C.1.

**Hybrid-Modality History Modeling.** Beyond traditional approaches, we further explore hybrid-modality history modeling strategies. However, as conversations unfold, models must repeatedly convert historical audio into text, which makes the reuse of audio KV-caches challenging. To address this issue, we compare three representative strategies: *I. Replace*: substitute the audio in the current round while directly reusing the KV-cache from other rounds; *II. Recomputation*: recompute all audio segments across rounds; *III. MoPE (modality-aware positional editing)*: a method inspired by He et al. (2024), which modifies positional encodings in a modality-aware manner. We report comparative performance under varying numbers of historical audio rounds (k). Details of MoPE are provided in Appendix C.2.

### 4.2.2 Experimental Results.

Table 3 compares accuracy and inference latency under different history-modeling strategies. We report results at three representative context lengths (0–2 min, 2–4 min, and 4–8 min) for the *Ultra Multi-Turn Dialogue* subset, and include a *Beyond-Semantic Dialogue* subset to probe information beyond pure semantics. For *Beyond-Semantic Dialogue*, because we use relatively small $k$ (the number of retained audio turns), hybrid methods are closely matched at longer ranges; thus, we only report the 0–2 min subset.

**Audio modeling alone struggles as history grows.** From 0–2 to 4–8 minutes, Audio-only (E1) accuracy plummets from 38.1% to 6.4%. By contrast, Text-only (E2) remains comparatively stable (45.8% → 41.2%; a 4.6-point drop). This disparity reflects token-budget pressure: audio is encoded as dense discrete units (multiple tokens per second) that rapidly saturate the context window, whereas text compresses history far more efficiently. As a result, modeling the entire history in audio is highly challenging and, under the same token budget, severely impairs long-context understanding. Even so, audio modeling remains valuable: in short contexts, adding a small amount of audio can slightly outperform text-only (e.g., 46.4% vs. 45.8% at 0–2 min), confirming that audio can convey richer information than text, even though its capacity for long-context modeling remains limited.

**Position editing enables efficient audio retention with favorable latency–accuracy trade-offs.** Naively substituting past audio (Substitution, E3/E6/E8) severely harms performance (e.g., at $k=1$ and 0–2 min: 27.8% vs. 46.4% for Re-computation, E4). By contrast, MoPE (ours) is nearly loss-less relative to recomputation across contexts and $k$: with $k=1$, it is within 0.6 points at 0–2 min (45.8% vs. 46.4%), and it consistently matches or surpasses Re-computation at longer contexts (e.g., 40.0% vs. 39.1% at 4–8 min for $k=3$, and 38.4% vs. 36.2% for $k=5$). These results confirm that modality-aware position editing reliably maintains the KV-cache of spoken dialogue without sac-

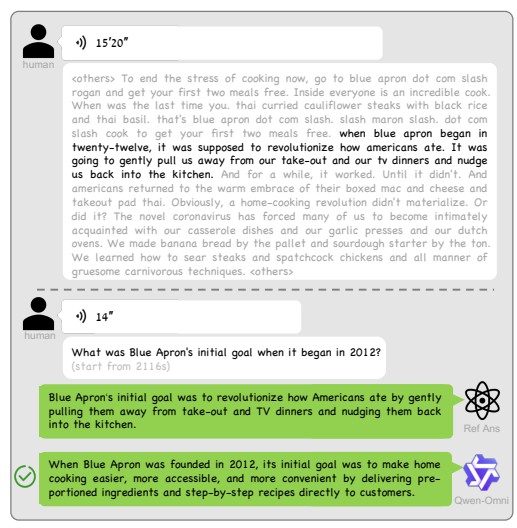

(a) Ultra Multi-Turn Dialogues.

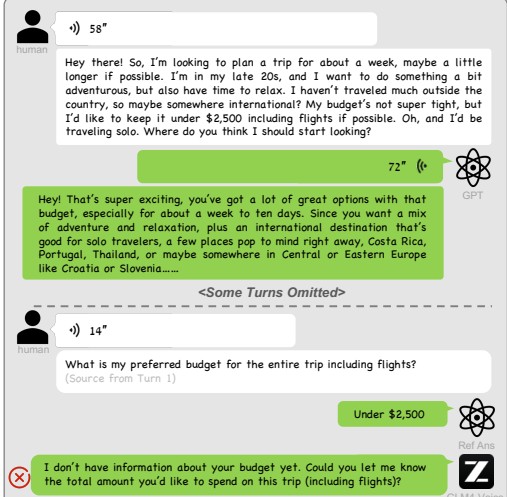

(b) Conversational Dialogue.

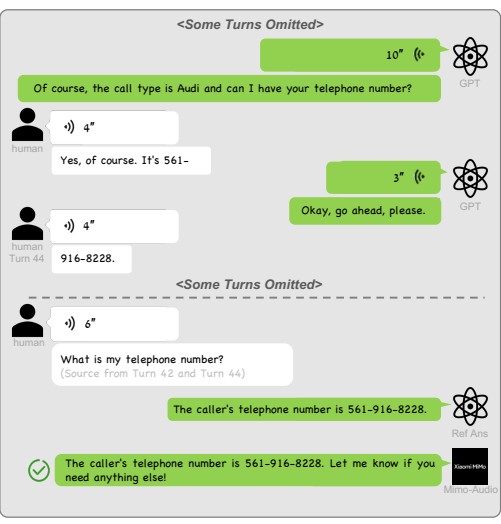

(c) Personal Monologues.

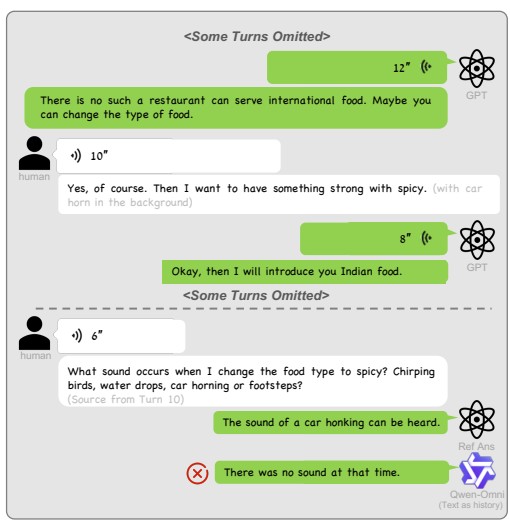

(d) Beyond-Semantic Dialogues.

Figure 4: Representative examples drawn from four subsets of VOX-INFINITY.

rificing accuracy. Beyond accuracy, MoPE offers a much better latency profile. As $k$ increases, Re-computation latency rises steeply and almost linearly (75 ms $\rightarrow$ 194 ms; +119 ms), whereas MoPE stays nearly flat (111 ms $\rightarrow$ 126 ms; +15 ms). With longer contexts and more retained audio turns, this efficiency advantage will become increasingly pronounced, while accuracy remains on par with—or even better than—recomputation.

In summary, audio carries essential acoustic cues beyond text, and MoPE effectively mitigates the latency burden of recomputation. By preserving recent acoustic information, maintaining long-range textual semantics, and achieving strong accuracy with significantly lower latency as $k$ grows, MoPE emerges as a promising modeling strategy for future long-context spoken language models.

## 4.3 LONG-CONTEXT SPOKEN DIALOGUE CASES IN VOX-INFINITY

To illustrate Vox-Infinity, Figure 4 presents one example from each subset. In each case, the light-gray region within the question indicates the answer's location, thereby specifying the amount of historical context required. The four datasets exhibit distinct characteristics, collectively covering

a broad spectrum of long-context speech-understanding scenarios. Due to context-length limitations, existing models (e.g., GLM-4-Voice) frequently fail on these tasks, as shown in Figure 4b. Moreover, in the absence of audio context, models such as Qwen2.5-Omni also struggle to capture information beyond semantics in Figure 4d. For additional examples, please refer to our demo page.

## 5 CONCLUSION

In this work, we present Vox-Infinity, a benchmark for evaluating long-context spoken language models. Vox-Infinity comprehensively covers diverse long-context understanding scenarios, including ultra multi-turn dialogues, personal monologues, and other spoken interaction settings. Through extensive experiments, we compare model performance across different history lengths and analyze the impact of various context modeling strategies. Furthermore, we explore approaches that balance long-range semantic understanding with the preservation of critical acoustic cues, aiming to provide more effective modeling of spoken dialogues.

## ETHICAL CONSIDERATIONS

Vox-Infinity was developed with careful attention to ethical and responsible research practices. All speech data used in this benchmark are publicly available, and no personally identifiable information is included. To further protect speaker privacy, the data are anonymized and used exclusively for research purposes.

We recognize that spoken dialogue corpora may reflect social, cultural, or gender biases, which could be inherited by models trained or evaluated on them. While Vox-Infinity is intended to advance research on long-context understanding, it should not be regarded as free of bias. We encourage future researchers to critically examine and mitigate potential harms that may arise from evaluation on this benchmark.

Finally, as large language and speech models become increasingly powerful, they also present risks of misuse, such as generating misleading content or enabling surveillance. Vox-Infinity is therefore released solely for academic and responsible industrial research, with the goal of fostering dialogue systems that are beneficial, transparent, and aligned with human values.

## REPRODUCIBILITY STATEMENT

We are committed to ensuring the reproducibility of our work. A portion of the dataset is already publicly available on our demo page, and the full release will be made available after paper acceptance. We provide detailed descriptions of the dataset construction process, evaluation protocols, and experimental settings in the main text and Appendix. In addition, the evaluation tools required for benchmarking will also be released upon acceptance to facilitate replication and future research.

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

## A    USE OF LLM

In this work, we employed large language models (LLMs) both for generating QA data and for conducting model evaluation. Moreover, as Vox-Infinity is proposed as a benchmark, we also report the performance of several LLMs on our dataset to provide reference baselines.

## B    PROMPT TEMPLATE

### B.1    PROMPT TEMPLATE FOR QA GENERATION.

We show the prompt template for QA generation in Figure 5. In addition to generating question–answer pairs, the process also records the source locations of the information needed for answering, ensuring traceability.

### B.2    EVALUATION PROMPT TEMPLATE

Figure 6 illustrates the GPT prompt template employed for model evaluation.

## C    HISTORY MODELING STRATEGIES

### C.1    TRADITIONAL MONO-MODALITY HISTORY MODELING

In spoken dialogue benchmarks, the choice of history representation $\mathcal{H}$ plays a critical role in model performance, particularly under long-context scenarios. We denote the audio history as $\mathcal{H}_a = \{x_1^{(a)}, x_2^{(a)}, \ldots, x_t^{(a)}\}$, where $x_i^{(a)}$ represents the raw acoustic signal of the $i$-th turn. The corresponding text history is defined as $\mathcal{H}_t = \{x_1^{(t)}, x_2^{(t)}, \ldots, x_t^{(t)}\}$, where each $x_i^{(t)}$ is obtained as the ASR transcript of its corresponding audio utterance $x_i^{(a)}$. Existing approaches typically adopt either audio-only or text-only history:

$$\hat{y_a} = \mathcal{D}(\mathcal{H}_a), \quad \hat{y_t} = \mathcal{D}(\mathcal{H}_t), \tag{1}$$

where $\mathcal{D}$ denotes the spoken dialogue model and $\hat{y}$ is the generated response. The audio-based form preserves prosody and speaker traits but is computationally expensive, while the text-based form is compact and LLM-friendly but discards acoustic cues such as intonation and background noise.

## C.2 Hybrid-Modality History Modeling

To retain richer acoustic information while constraining the overall growth of history length, we propose a hybrid history representation $\mathcal{H}_h = \{x_1^{(t)}, \ldots, x_{t-k}^{(t)}, x_{t-k+1}^{(a)}, \ldots, x_t^{(a)}\}$, where earlier turns ($1 \leq i \leq t - k$) are stored in text form and the most recent $k$ turns ($t - k + 1 \leq i \leq t$) are preserved in audio form. This design keeps recent context acoustically rich while compressing older history into transcripts.

Since spoken dialogue models process conversations sequentially, the dialogue length $t$ of $\mathcal{H}_h$ inevitably grows over time. A challenge arises when editing the KV cache: once a new audio turn $x_{t+1}^{(a)}$ is added, the previous boundary turn $x_{t-k+1}^{(a)}$ should be converted into $x_{t-k+1}^{(t)}$, and all cached keys with $i \geq t - k + 1$ must correctly reflect their new positions; otherwise, the attention scores will become inconsistent.

To address this non-alignment challenge, we introduce **Mo**dality-Aware **P**ositional **E**diting (MoPE), which corrects cached keys by removing their outdated rotary positional encodings and reapplying the correct ones. Standard rotary positional encoding (RoPE) (Su et al., 2024), the default positional embedding in most LLMs, encodes position $j$ directly into cached keys via the rotary matrix $R_j$. For a cached key $k_{j'}^l$ at layer $l$ and original position $j' \geq t - k + 1$, the edited version is computed as

$$k_{j'}^{\text{edit},l} = R_m \, k_{j'}^l, \tag{2}$$

where $m$ denotes the relative index offset introduced when converting the boundary turn $x_{t-k+1}^{(a)}$ into its text form $x_{t-k+1}^{(t)}$, and $R_m$ is the rotary matrix parameterized by this offset. This transformation is valid because RoPE exhibits a *group property*: the rotary matrices compose additively with respect to position offsets, i.e., $R_{\Delta_1} R_{\Delta_2} = R_{\Delta_1 + \Delta_2}$ and $R_{-\Delta} = R_\Delta^{-1}$. Therefore, applying $R_m$ to a cached key originally encoded at $j'$ is equivalent to repositioning it from $j'$ to $j' + m$, ensuring positional consistency without the need for full re-encoding. The updated KV cache is then reconstructed as:

$$\begin{aligned}
K^{\text{edit}} &= \text{Concat}\big(K_{1:t-k}^{(t)}, \ K_{t-k+1}^{\text{edit},(t)}, \ K_{t-k+1+m:t+1}^{\text{edit},(a)}\big), \\
V^{\text{edit}} &= \text{Concat}\big(V_{1:t-k}^{(t)}, \ V_{t-k+1}^{\text{edit},(t)}, \ V_{t-k+1+m:t+1}^{(a)}\big),
\end{aligned} \tag{3}$$

where $K$ and $V$ denote the key and value caches, respectively. The red-highlighted terms indicate the updated entries after converting the boundary turn into text and appending the new audio tokens.

Unlike full recomputation, MoPE introduces only a single rotary transformation, resulting in negligible overhead. Applied to our **hybrid-modality history**, MoPE guarantees that the transition from text to audio history remains positionally consistent after edits, thereby enabling seamless reuse of cached representations across modalities.

## D Consistency between human evaluation and Gpt-Metric.

To assess the consistency between the GPT-based metric and human evaluation, we recruited five professional annotators to label the model outputs, as presented in Table 5. The final human judgment was determined through majority voting. All experiments results were conducted using Qwen2.5-Omni. The results show a strong alignment between the GPT-based metric and human evaluations, indicating that the GPT-based metric is reliable and suitable for use as a valid evaluation measure.

## E Memory analysis under different history modeling strategies.

Since the core architectures of the evaluated models are similar, the primary differences in memory usage stem from their context-storage footprint. We here provide an analysis of how GPU memory consumption scales with the number of audio turns retained in the context, offering a clearer picture of the resulting memory behavior. As shown in Table 4, the data itself has little impact on memory usage, and the variation in GPU consumption across different history-modeling strategies is almost negligible.

Table 4: Memory analysis under different history modeling strategies.

| Method | Audio Turns | Mem (GB) |
|---|---|---|
| audio-only | $n$A | 23.358 |
| text-only | $n$T | 22.612 |
| Substitution | $(n-1)$T+1A | 22.777 |
| recompute | $(n-1)$T+1A | 22.875 |
| mope | $(n-1)$T+1A | 22.732 |
| recompute | $(n-3)$T+3A | 23.374 |
| mope | $(n-3)$T+3A | 22.824 |
| recompute | $(n-5)$T+5A | 23.432 |
| mope | $(n-5)$T+5A | 22.936 |

Table 5: Consistency analysis between the GPT-based metric and human evaluation. All results are generated using Qwen2.5-Omni and Mimo-audio.

| Method | Ultra Multi-Turn | | | Conversational | | | | Personal Monologues | | | | Beyond-Semantic | | |
|---|---|---|---|---|---|---|---|---|---|---|---|---|---|---|
| | 0-2m | 2-4m | 4-8m | 0-2m | 2-4m | 4-8m | 8m+ | 0-2m | 2-4m | 4-8m | 8m+ | 0-2m | 2-4m | 4-8m |
| *Qwen2.5-Omni* | | | | | | | | | | | | | | |
| GPT-eval | 60.2 | 58.9 | 57.4 | 88.5 | 77.8 | 85.3 | 79.6 | 83.2 | 50.0 | 52.6 | 49.0 | 17.3 | 15.9 | 15.7 |
| Human-eval | 60.8 | 59.3 | 56.9 | 88.6 | 77.8 | 85.2 | 79.6 | 83.5 | 52.2 | 52.8 | 48.6 | 17.2 | 16.0 | 15.7 |
| *Mimo-Audio* | | | | | | | | | | | | | | |
| GPT-eval | 35.7 | 30.6 | 25.9 | 90.4 | 94.4 | 85.3 | 86.6 | 78.1 | 41.7 | 15.8 | 9.8 | 22.1 | 20.4 | 21.7 |
| Human-eval | 35.2 | 31.6 | 25.3 | 91.2 | 94.1 | 85.7 | 86.1 | 78.5 | 41.1 | 15.5 | 9.5 | 22.4 | 20.1 | 21.2 |

## LIMITATION

In the domain of long-context modeling, various tasks exist, such as full-document understanding, discourse-level reasoning, and context-based retrieval. In this work, however, we restrict our focus to understanding tasks. The reason is that many other tasks are essentially text-centric and primarily reflect the inherent capabilities of the underlying language model. In contrast, our contribution lies in examining how history information should be represented and modeled—a challenge that traditional text-only LLMs do not encounter. Therefore, this paper centers its discussion on this specific but crucial problem.

When evaluating spoken dialogue, acoustic features like emotion and speaker identity carry a high risk of semantic leakage (e.g., inferring emotion from text alone, or deducing speaker identity via textual logic and recent speech). To prevent this in our beyond-semantic dialogue benchmark, we opted to use semantically independent sound events as the acoustic information. Although this choice guarantees strict evaluative purity, it currently lacks the assessment of other acoustic dimensions, such as emotion. We plan to address this limitation in future work by integrating this information to achieve a more comprehensive evaluation.

You are a researcher studying large speech models and their ability to **retain and recall information from historical voice dialogues**. Your goal is to evaluate whether a model can answer user questions based on previous dialogues after many conversational turns. To do this:

- You will receive **transcriptions** of several audio samples.
- Each transcription contains the spoken content (speech only).
- Use the transcriptions to reconstruct the conversation and gather information.

**Example input format:**
```
{
  "Audio 0": { "Transcription": "hi , my name is john . i'm looking for info in cambridge ." },
  ...
  "Audio 39": { "Transcription": "okay , thank you so much for calling . goodbye ." }
}
```

**Your task:** Generate questions and answers based on the dialogue.
**Requirements:**

1. Questions must relate to the audio content.
2. Answers should be **short, clear, and concise** for easy evaluation.
3. Questions should require **reasoning, world knowledge, and information extraction**, not trivial restatement.
4. Do **not** refer to the transcription explicitly; assume the model only "hears" the audio.
5. Each answer should reflect the **final resolved information** (not partial mid-conversation states).
6. Avoid ambiguity: every question should have a **single, clear answer**.
7. Add a "Source" field citing the audio(s) that provide the answer.
8. Context length control:
    - First question comes from the most recent stage, the second from the next, and so on.
    - If an answer spans multiple audios, ask at the stage where the information is complete.
    - As turns increase, "Source" audios should progressively move earlier.

**Output format:**
```
{
  "QA-Pair 1": {
    "Question": "What are the open hours of the theatre?",
    "Answer": "The theatre is open all the time.",
    "Source": { "Audio 35": "okay , actually they are open all the time ." }
  }
  ...
}
```

Figure 5: Prompt template for QA generation.

**Prompt Template**

You are an AI assistant tasked with judging the correctness of a model's response to a multiple-choice question. You are given the question, its answer options, the correct answer, and the model's output. Your evaluation must follow these rules:

- If the model's output matches the correct answer, output "Correct".
- If the model's output matches one of the options but not the correct answer, output "Incorrect".
- If the model's output does not match any of the options, output "Unknown".

Your output must be a single word chosen strictly from: {Correct, Incorrect, Unknown}.

**Question:** {question}
**Options:** {options}
**Correct Answer:** {correct_answer}
**Model Output:** {model_output}

Figure 6: Standardized prompt template used for QA evaluation.

