# OpenReview forum: "Vox-Infinity: Benchmarking the Limits of Long-Context Spoken Language Models"
_ICLR.cc/2026/Conference — Submitted to ICLR 2026_

### Official Review · Reviewer_inoe · 2025-10-26

**Soundness:** 3
**Presentation:** 3
**Contribution:** 3
**Rating:** 6
**Confidence:** 4

**Summary:**

This paper introduces a benchmark for evaluating long-context understanding in spoken language models. It scales dialogue history by turn count and duration, includes answer provenance annotations, and covers diverse scenarios. The study compares text-, audio-, and hybrid-history strategies, highlighting trade-offs and proposing Modality-Aware Positional Editing (MoPE) for efficient context modeling.

**Strengths:**

1. The paper is highly original in framing long-context understanding as a core challenge in spoken language modeling, an underexplored domain compared to text-based benchmarks.

2. The paper demonstrates strong methodological quality through detailed dataset construction, human validation, and systematic model benchmarking. I

3. The paper is well-organized, easy to follow, and supported by clear figures, tables, and examples.

**Weaknesses:**

1.  Evaluation is limited to a few models; broader inclusion of emerging audio-LLMs could better validate Vox-Infinity’s scalability and generality.

2. The benchmark focuses mainly on QA tasks; adding generative or reasoning evaluations would strengthen coverage of long-context language understanding.

3. MoPE’s algorithmic details lack ablation on parameter sensitivity; reporting computational cost variations would clarify its efficiency and reproducibility.

**Questions:**

none

---

> ### Author Response · Authors · 2025-11-21
> **Rebuttal to Reviewer inoe**
>
> We sincerely appreciate your recognition of the motivation behind our benchmark, as well as your positive comments on our data construction process and experimental analysis. Please find our detailed responses below.
>
> ### **Q1. Limited LALM models**
>
> Thank you for your valuable feedback. In the revised version, we have incorporated additional audio-language models (LALMs), including Qwen-2 Audio, Step-Audio, and GPT-Realtime, to enable a more comprehensive and systematic evaluation. Before the rebuttal deadline, we will continue adding results from more LALMs to further enrich our assessment.
>
> ### **Q2. Advice on adding generative or reasoning evaluations**
>
> Thank you for your insightful suggestion. We agree that QA, generative, and reasoning tasks constitute core paradigms in many text-based benchmarks. Our current benchmark includes two task types:
>
> - **Generative open-ended QA** (Ultra Multi-turn Dialogues and Personal Monologues), and
> - **Multiple-choice QA** (Conversational Dialogue and Beyond-Semantic),
> both of which focus on querying specific details from the dialogue history.
>
> Regarding reasoning-oriented tasks, these typically require higher-level inference based on basic comprehension. While such tasks may partially reflect long-context abilities, they do not isolate the capability of modeling long-context history itself. Instead, they tend to conflate context retention with multi-step reasoning skill. We believe this represents an exciting direction for future extensions—evaluating long-context joint comprehension and reasoning capabilities—and we look forward to exploring these tasks in future work.
>
> ### **Q3. The ablation of MoPE**
>
> As described in line 741, the only parameter in MoPE is the number of preserved audio turns (*k*). We performed a fine-grained ablation study on this parameter, and the results are presented in **Table 3** and **Section 4.3**. This analysis reveals the trade-off between inference speed and performance under different context lengths.
>
>
> Once again, we sincerely appreciate your constructive feedback and your recognition of our contributions. We hope our responses sufficiently address your concerns, and we would be glad to continue the discussion if you have any additional questions.

---

### Official Review · Reviewer_EdSM · 2025-10-27

**Soundness:** 2
**Presentation:** 2
**Contribution:** 3
**Rating:** 4
**Confidence:** 4

**Summary:**

This paper presents a new benchmark, Vox-Infinity, benchmarking long context for spoken language models. It mainly compares on three models: GLM-4 Voice, Mimo-Audio and Qwen2.5-Omni. The evaluation reveals that text-based history modeling remains more stable as context length increases, while audio-based history preserves crucial acoustic cues but degrades rapidly with longer contexts. The austhors also propose a new approach (MoPE) to handle this problem.

**Strengths:**

- First benchmark specifically for evaluating long-context spoken dialogue.
- 154 hours of audio covering four scenarios.
- Propose Modality-Aware Positional Editing, a way to modify positional encodings.

**Weaknesses:**

- Only three models evaluated (GLM-4-Voice, Qwen2.5-Omni, Mimo-Audio) while many other spoken dialogue systems exist.
- Section 4.2 dedicates significant space to a relatively incremental position encoding method, not within the scope of a benchmark paper.
- Relying solely on GPT-4o for accuracy assessment without human evaluation baseline or inter-rater agreement is problematic.
- Table 2 is dense with too many columns making trends hard to identify.
- While latency figures are provided, the paper lacks a discussion on memory requirements.

**Questions:**

- Given the paper's reliance on GPT-4o for evaluation , have the authors conducted any human validation studies or inter-rater agreement checks to confirm the reliability of these automated judgments?
- Regarding the "Conversational Dialogues" subset, could the authors specify which Text-to-Speech (TTS) model was used for audio synthesis and what methods were employed to manage speaker identity?
- The paper provides valuable latency metrics, but could the authors also report the memory requirements (e.g., VRAM) for running the benchmark? This information is critical for assessing the total computational cost.
- Could the authors elaborate on the decision to dedicate a significant portion of the analysis (Section 4.2) to the new MoPE method?
- The evaluation is limited to three models. Consider adding AudioFlamingo3, Qwen-2 Audio, Step-Audio, UltraVox, GPT-Realtime etc.
- Would the authors consider providing an aggregate performance metric or reorganizing Table 2? The current dense format makes it difficult to quickly identify performance trends.

---

> ### Author Response · Authors · 2025-11-21
> **Rebuttal to Reviewer EdSM**
>
> Thank you for your thoughtful and constructive comments. We address each point in detail below.
>
> ### **Q1. Limited LALM models**
>
> Thank you for your valuable feedback. In the revised version, we have incorporated additional audio-language models (LALMs), including Qwen-2 Audio, Step-Audio, and GPT-Realtime, to enable a more comprehensive and systematic evaluation. Before the rebuttal deadline, we will continue adding results from more LALMs to further enrich our assessment.
>
> ### **Q2: Why introduce MoPE?**
>
> A core challenge for LALMs in long-context understanding is balancing **rich acoustic preservation** with **manageable historical length**. Audio-as-history modeling typically requires **12.5–25 tokens per second**, while text requires only **2–3 tokens**. Therefore, the strategy for dialogue history preservation (i.e., how many turns to keep in audio form) is a critical issue when discussing long-context audio modeling.
>
> In Section 4.2, we analyze how different history-modeling methods affect long-context performance. Existing LALMs mostly fall into two extremes:
>
> - **Audio-only history modeling** (e.g., GLM-4-Voice, MIMO-Audio), and
> - **Text-only history modeling** (e.g., Audio-Flamingo, Step-Audio, SLAM-Omni).
>
> Naturally, this raises an important question: *Is there a practical trade-off between these two extremes that allows us to preserve more acoustic information without excessively limiting context length?*
>
> To explore this, we introduce MoPE as a *middle-ground strategy* between audio-only and text-only history modeling. We would like to emphasize that MoPE is **not** presented as a main contribution of our work; rather, it helps us conduct a more complete and systematic comparison of history-preservation approaches.
>
> ### **Q3: Consistency between GPT-based metrics and human evaluation**
>
> Our tasks focus on QA with relatively deterministic answers, which makes GPT-based scoring reasonably stable and reliable. To further validate this, we additionally conducted human evaluation on Qwen2.5-Omni and observed strong alignment between GPT-based metrics and human judgments. UMTD for ultra multi-turn dialogue, CD for conversational dialogue, PM for personal monologue, BSD for Beyond-semantic dialogues.
>
> |  | **UMTD** | **UMTD** | **UMTD** | **CD** | **CD** | **CD** | **CD** | **PM** | **PM** | **PM** | **PM** | **BSD** | **BSD** | **BSD** |
> | --- | --- | --- | --- | --- | --- | --- | --- | --- | --- | --- | --- | --- | --- | --- |
> | Method | 0-2m | 2-4m | 4-8m | 0-2m | 2-4m | 4-8m | 8m+ | 0-2m | 2-4m | 4-8m | 8m+ | 0-2m | 2-4m | 4-8m |
> | GPT-eval | 60.2 | 58.9 | 57.4 | 88.5 | 77.8 | 85.3 | 79.6 | 83.2 | 50.0 | 52.6 | 49.0 | 17.3 | 15.9 | 15.7 |
> | Human-eval | 60.8 | 59.3 | 56.9 | 88.6 | 77.8 | 85.2 | 79.6 | 83.5 | 52.2 | 52.8 | 48.6 | 17.2 | 16.0 | 15.7 |
>
> ### **Q4: The large number of columns in Table 2**
>
> Thank you for pointing this out. In the updated version, we have added a mean column to improve readability and moved the detailed breakdown—now grouped by audio duration—to the appendix. This keeps the main table clear and concise while still allowing readers to examine long-context performance trends in detail. We welcome any additional suggestions you may have regarding further improvements in presentation.
>
> ### **Q5: Memory requirements**
>
> Since the core architectures of the evaluated models are similar, the primary differences in memory usage stem from their context-storage footprint. We here provide an analysis of how GPU memory consumption scales with the number of audio turns retained in the context, offering a clearer picture of the resulting memory behavior. As shown by the experimental results, the data itself has little impact on memory usage, and the variation in GPU consumption across different history-modeling strategies is almost negligible.
>
> | Method | Audio Turns | mem (GB) |
> | --- | --- | --- |
> | audio-only | k=+∞ | 23.358 |
> | text-only | k=0 | 22.612 |
> | Substitution | k=1 | 22.777 |
> | recompute | k=1 | 22.875 |
> | mope | k=1 | 22.732 |
> | recompute | k=3 | 23.374 |
> | mope | k=3 | 22.824 |
> | recompute | k=5 | 23.432 |
> | mope | k=5 | 22.936 |
>
> Once again, we sincerely appreciate your valuable feedback. Your comments have helped us improve both the clarity and completeness of our work, and we would be glad to discuss any further suggestions you may have.

---

> > ### Comment · Reviewer_EdSM · 2025-11-27
> >
> > I thank the authors for their response. The authors reply using a different numbering (apparently aligned with my listed weaknesses rather than my explicit questions), so I follow their ordering.
> >
> > Q1, Q2 and Q4 have been addressed. For Q3, the new human evaluation is limited to a single model and no inter-rater agreement statistics are reported. For Q5, the memory analysis appears only in the rebuttal and is not yet clearly integrated into the paper.
> >
> > My original question about the TTS system and speaker identity remains unaddressed. Given these remaining issues and the overall scope and presentation of the work, I would like to maintain my original score of 4.

---

> ### Author Response · Authors · 2025-11-27
>
> Thank you for your thoughtful feedback and for acknowledging that we have addressed Q1, Q2, and Q4. Please allow us to further respond to the remaining questions.
>
>
>
> ### **Q3: Consistency between human evaluation and the GPT-based metric**
>
> To obtain human evaluation scores, we recruited five professional annotators to label the model outputs, and the final human judgment was determined through majority voting. For reasons of evaluation fairness and scalability, it is not feasible for us to conduct human annotation across all metrics. Our primary goal is to demonstrate the consistency between human judgments and the GPT-based metric.
>
> Following the practices of prior benchmarks [1], we believe that establishing strong consistency on a representative model is sufficient to validate the reliability of the GPT-based metric. Nevertheless, to meet your expectations, we will make every effort to include additional human evaluation results before the rebuttal deadline to further reinforce this conclusion.
>
>
>
> ### **Q5: Memory analysis**
>
> Thank you again for your valuable suggestion. We have added the memory analysis results to Appendix E. Your feedback has helped us improve the completeness of our work.
>
>
>
> ### **Q6: TTS details and speaker identity for conversational dialogue subset**
>
> Thank you for raising this question, and we apologize for inadvertently overlooking it earlier. To construct the dataset, we used CosyVoice2-0.5B for speech synthesis due to its strong expressiveness and high-quality generation. To ensure consistent speaker timbre, we followed prior work [2] and applied a speaker verification [3]. All synthesized audio was further validated using whisper to ensure correct pronunciation. We have added the detailed information to Section 3.1 of the main paper.
>
>
>
> Thank you for all your suggestions, which have helped us improve the overall scope and presentation of the paper. We hope our responses have addressed your remaining concerns. If you have any further questions, we would be grateful for the opportunity to discuss them. **We sincerely hope that our efforts positively influence your evaluation of our work.** Thank you again for your careful and constructive feedback.
>
> [1] Hallusionbench: an advanced diagnostic
> suite for entangled language hallucination and visual illusion in large vision-language models. CVPR 2024
>
> [2] Voxdialogue: Can spoken dialogue systems understand
> information beyond words? ICLR 2025
>
> [3] https://huggingface.co/pyannote/speaker-diarization-3.1

---

### Official Review · Reviewer_LqdZ · 2025-10-30

**Soundness:** 3
**Presentation:** 3
**Contribution:** 3
**Rating:** 8
**Confidence:** 4

**Summary:**

This paper introduces the very first benchmark specifically designed to evaluate the limits of long-context understanding in spoken language models. The benchmark systematically scales context length across two dimensions: dialogue turns and turn duration, and includes four diverse scenarios: Ultra Multi-turn Dialogue, Conversational Dialogue, Personal Monologue, and Beyond-Semantic Dialogue. The paper also provides a comparison of text-only, audio-only, and hybrid history modeling strategies, demonstrating key trade-offs in semantic retention versus acoustic preservation.

**Strengths:**

1.	The motivation behind the paper is strong, addressing the need for a dedicated, rigorous benchmark for long-context spoken language models.
2.	The paper itself is clearly written.
3.	The analyses in the paper are well-designed.

**Weaknesses:**

1.	Limited numbers of LALMs, as only 3 open-source models are provided.

**Questions:**

1.	Why, in Table 1, Length Distribution for Conversational Dialogue presents 0-2min and 2-4min, but in Table 2, numbers for 4-8min and 8min+ are also provided?
2.	What does “Value” in Figure 3(a) indicate?

---

> ### Author Response · Authors · 2025-11-21
> **Rebuttal to Reviewer LqdZ**
>
> Thank you for your recognition of our benchmark motivation, writing, and analysis.
>
> **Q1. Limited LALM models**
> A1: Thank you for your valuable feedback. In the revised version, we have incorporated additional audio-language models (LALMs), including Qwen-2 Audio, Step-Audio, and GPT-Realtime, to enable a more comprehensive and systematic evaluation. Before the rebuttal deadline, we will continue adding results from more LALMs to further enrich our assessment.
>
> **Q2: Inconsistency between Table 1 and Table 2**
>
> A2: We apologize for the typographical errors. The length distribution of Conversational Dialogue in Table 1 should be 0–2 min, 2–4 min, 4–8 min, and 8 min+.
>
> **Q3: Meaning of the values in Table 3(a)**
>
> A3:The values represent accuracy for different context lengths. We have clarified this definition in the latest version of the paper.
>
> Once again, we sincerely appreciate your constructive feedback and recognition of our work. We hope our responses address your concerns, and we would be glad to continue the discussion if you have any further questions.

---

### Official Review · Reviewer_cPLr · 2025-10-31

**Soundness:** 2
**Presentation:** 3
**Contribution:** 2
**Rating:** 4
**Confidence:** 4

**Summary:**

The paper proposes a new benchmark for analyzing spoken language models (SLMs) on long-context tasks. Existing long-context benchmarks typically cover either short scenarios (30 seconds – 1 minute) or extremely long ones (around 1 hour). However, these settings are either too short or too long to reflect realistic conversation lengths. This work fills the gap by introducing benchmark clips ranging from less than 2 minutes to 8 minutes.

The authors design four dialogue scenarios and categorize the testing clips into four types:
(a) Ultra Multi-turn Dialogue – frequent back-and-forth exchanges involving requirement verification and detail clarification.
(b) Conversational Dialogue – extended, casual interactions emphasizing sustained conversation and advice-giving.
(c) Personal Monologue – single-speaker, long-form speech such as lectures or presentations.
(d) Beyond-Semantics Dialogue – conversations enriched with acoustic information (e.g., environmental sounds, sound effects, or music) beyond pure semantics.

The study shows that existing SLMs perform well when using text-based context histories, with little degradation from short to long contexts. However, when audio is used as the context, performance degrades significantly in long-context scenarios. Interestingly, the authors also demonstrate that audio information is essential, as audio-based context modeling outperforms text-based modeling in the Beyond-Semantics Dialogue category.

Finally, they propose a hybrid context modeling approach that uses text for distant history and audio for recent history. However, the effectiveness of this hybrid method is not well established by the presented experiments.

**Strengths:**

- A clear gap in evaluating long-context modeling in SLMs
- Include the "Beyond-Semantics Dialogue" category that gauge the essentialness of audio modeling.

**Weaknesses:**

### Review Comment

- Given that this is a speech benchmark, I would expect most of the testing scenarios to require the model to rely on **paralinguistic** or **acoustic information** in the context history. However, only one category currently addresses the importance of acoustic information, and it focuses on **sounds and music** rather than other rich aspects of speech beyond verbal content (e.g., **speaker identity**, **emotion**, **tone**). This design makes the benchmark easily solvable using **pure text-based context modeling**. As shown by the experiments in the paper, none of the context modeling methods involving audio significantly outperform pure text modeling, including the proposed hybrid approach. Audio-based methods merely catch up with text-based modeling, but at a significantly higher **latency cost**. Ideally, this benchmark should advocate for audio-based context modeling, yet the results tell the **opposite story**.

- The effectiveness of the proposed **hybrid modeling** is also questionable. The paper only reports results on the *“Ultra Multi-Turn Dialogue”* category, where acoustic signals are largely unnecessary. What about the *“Beyond-Semantics Dialogue”* category? How does the hybrid method compare to the pure audio-based method in that case? Since the hybrid method only models acoustics for recent history, it is very likely to outperform the pure audio-based method in *“Ultra Multi-Turn Dialogue”*, but underperform it in *“Beyond-Semantics Dialogue”*. If this is the case, the contribution of the proposed hybrid method becomes **unclear**.

---

### Summary of Weaknesses

1. The benchmark does **not fully consider the rich acoustic information** in speech to design tasks that require the SLM to *“listen to”* the context rather than merely *“look at”* it.
2. The paper does **not propose an effective context modeling method** that surpasses text-based approaches by additionally modeling context audio. This limitation may stem from weaknesses in either the **benchmark design** or the **method design**.

**Questions:**

No

---

> ### Author Response · Authors · 2025-11-21
> **Rebuttal to Reviewer cPLr (1/n)**
>
> Thank you very much for your positive assessment of the motivation and experimental design of our work. Please allow us to provide more detailed responses to your questions.
>
> ### **W1-1: Why does the acoustic evaluation focus only on environmental sounds, rather than paralinguistic cues such as emotion or speaking rate?**
>
> **A1-1:** Thank you for raising this important question. We have also thought carefully about this issue, and our considerations are as follows.
>
> Acoustic information in speech can generally be categorized into three types: speaker-related attributes, paralinguistic attributes, and environmental attributes.
>
> 1. **Speaker-related attributes.**
> In spoken dialogue—arguably the most common long-context speech-understanding scenario—the speaker identity typically remains fixed. Models can therefore capture all necessary speaker cues from the most recent turns, making this type of attribute unsuitable for evaluating long-context audio-understanding ability.
> 2. **Paralinguistic information (e.g., speaking rate, volume, emotion).**
> Speaking rate and volume are inherently *relative*, lacking an absolute scale, so they are rarely used as core features for evaluating audio understanding. Emotion, in turn, is highly correlated with textual content, meaning that performance may not purely reflect acoustic understanding. This introduces confounds that make it difficult to determine whether models rely on acoustic cues or semantic inference, thereby weakening evaluation reliability.
> 3. **Environmental acoustic events.**
> Environmental sounds are independent of semantic content and therefore directly can be used to test whether a model truly understand acoustic information, rather than relying on text-based reasoning.
>
> For these reasons, we chose to focus on environmental acoustic events for long-context evaluation to LALMs.
>
> ### **W1-2: “This benchmark should advocate for audio-based context modeling.”**
>
> **A1-2:** First, we would like to emphasize that **a benchmark should aim to provide fair comparisons and analyses, rather than advocating for a particular modeling paradigm**. As discussed in the main text, text-only history modeling and audio-only history modeling each have their own strengths. Our goal is therefore to compare these two approaches comprehensively and fairly across different scenarios.
>
> To reiterate the rationale behind the four subsets in our benchmark:
>
> - Conversational Dialogue, Ultra Multi-turn Dialogues, and Personal Monologues correspond to real-world semantic speech-understanding settings and primarily assess *semantic* long-context comprehension. In these scenarios, the text-as-history paradigm (as used in SLAM-Omni and Step2-Audio) remains the most effective means of preserving semantic information over long contexts.
> - In contrast, the Beyond-Semantic subset is specifically designed to test long-context behavior in *acoustic* understanding that cannot be solved using textual cues alone. In this scenario, audio-as-history modeling strategy demonstrates clear advantages.

---

> > ### Author Response · Authors · 2025-11-21
> > **Rebuttal to Reviewer cPLr (2/n,n=2)**
> >
> > ### **W2: Why do we evaluate on Ultra Multi-turn Dialogues?**
> >
> > In Table 3, our primary aim is to examine how different history-modeling strategies (i.e., preserving previous turns in audio form) affect long-term memory. Therefore, we evaluate using the semantic-focused Ultra Multi-turn Dialogues subset. If we were to use the Beyond-Semantic subset, any turns not preserved in audio would become unanswerable, which would prevent meaningful comparison of long-context modeling capabilities.
> >
> > To better meet your expectations, we also evaluated accuracy under different numbers of retained turns $k$ using Mimo-Audio. Because acoustic information has a substantial impact on the beyond-semantic dialogue subset, we provide a fine-grained analysis of how performance varies with different answer-turn indices ($t$). Here, $t = m$ indicates that answering the question requires referring to the $m$-th most recent turn. The results show that, without introducing audio, the model fails completely on acoustically related questions. As the number of turns increases, the model exhibits clear limitations in long-context modeling, leading to degraded understanding of acoustic cues from earlier turns. In contrast, as long as MoPE retains the audio for the required turns, the model can effectively interpret the corresponding acoustic information, achieving performance nearly on par with the audio-only approach. These results will be included in the revised version of our paper.
> >
> > | **Method** | t=1 | t=3 | t=5 |
> > | --- | --- | --- | --- |
> > | Audio-only | 28.7% | 25.3% | 22.3% |
> > | Text-only | 6.9% | 6.8% | 7.0% |
> > | Re-computation(k=1) | 28.6% | 6.9% | 7.1% |
> > | Mope(k=1) | 28.5% | 7.0% | 6.9% |
> > | Re-computation(k=3) | 28.7% | 25.3% | 7.0% |
> > | Mope(k=3) | 28.4% | 25.1% | 7.2% |
> > | Re-computation(k=5) | 28.7% | 25.2% | 22.4% |
> > | Mope(k=5) | 28.6% | 25.0% | 21.9% |
> >
> > Once again, we sincerely appreciate your constructive feedback and your recognition of our contributions. We hope our responses address your concerns, and we would be happy to continue the discussion if you have further questions.

---

> > > ### Comment · Reviewer_cPLr · 2025-11-25
> > > **Reply to the author**
> > >
> > > > Speaker-related attributes. In spoken dialogue—arguably the most common long-context speech-understanding scenario—the speaker identity typically remains fixed. Models can therefore capture all necessary speaker cues from the most recent turns, making this type of attribute unsuitable for evaluating long-context audio-understanding ability.
> > >
> > > This is not true. In a meeting or multi-party conversation scenarios, the dialogue is long and the speaker turns are dynamic. The response from the author is not convincing. They lack the study on the speaker change. It does not mean this scenario does not exist or is not important.
> > >
> > > > Paralinguistic information (e.g., speaking rate, volume, emotion). Speaking rate and volume are inherently relative, lacking an absolute scale, so they are rarely used as core features for evaluating audio understanding. Emotion, in turn, is highly correlated with textual content, meaning that performance may not purely reflect acoustic understanding. This introduces confounds that make it difficult to determine whether models rely on acoustic cues or semantic inference, thereby weakening evaluation reliability.
> > >
> > > I agree with the speaking rate and volume part, but disagree with the emotion part. In realistic dialogue, speech can be expressive and emotional to convey the true intent, and that is a key difference from text-based dialogue. Modeling such realistic and complex scenarios helps foster more human-like and empathetic spoken agents. The job of a benchmark should be determining the problem to be solved and collecting the most realistic data for it, instead of neglecting a certain aspect simply because it is harder to analyze.
> > >
> > > Of course, you can claim that since the problem is too hard for analysis, we simplify it and do not discuss speaker and emotional changes for this spoken dialogue dataset. However, this action also leads to limited contribution and significance. In that case, the benchmark dataset simply becomes longer in utterance compared to the existing ones. I acknowledge that this is still a valid contribution, but I do not think it is significant enough for an ICLR submission.
> > >
> > > > W1-2: “This benchmark should advocate for audio-based context modeling.”
> > >
> > > I agree with the statement that “a benchmark should aim to provide fair comparisons and analyses, rather than advocating for a particular modeling paradigm.” However, I would like to clarify my point. In my previous review, my argument is that a realistic speech benchmark should include speaker and emotional dynamics, which are not captured by pure text-based dialogue. As a result, audio-based context modeling should, in principle, be more effective in understanding such acoustic changes.
> > >
> > > In this work, the authors largely confine their problem formulation to semantic-only dialogue. I would not say this is “more fair.” By neglecting speaker and emotional dynamics, the benchmark inherently favors pure text-based context modeling. A truly “fair” benchmark should comprehensively and realistically reflect a broad spectrum of scenarios. When only semantic-only dialogues are presented, it is natural that pure text-based context modeling performs better. But is that result truly fair and truly reflective of real-world spoken interaction? Without studying the richer acoustic variations, we cannot know.
> > >
> > > This is the major gap that I believe the paper should address to be more complete for an ICLR submission.

---

> > > > ### Comment · Reviewer_cPLr · 2025-11-25
> > > > **Summarizing the discussion**
> > > >
> > > > In conclusion, the authors present a long spoken dialogue dataset. A long spoken dialogue dataset is not a novel problem, as it has been discussed previously. Existing long-context benchmarks typically cover either short scenarios (30 seconds to 1 minute) or extremely long ones (around 1 hour). This work fills the gap by introducing benchmark clips ranging from less than 2 minutes to 8 minutes.
> > > >
> > > > The dataset is mostly semantic-focused; that is, in most cases, only the spoken content in the dialogue context matters, and the paralinguistic and speaker identities do not matter. In my point of view, speaker identity and emotional tone are crucial in realistic conversation and should be discussed when comparing text-based and audio-based context-modeling methods. The paper lacks such a study, which I consider a major weakness leading to a rejection. However, I am still open to seeing it accepted, as the remaining parts are complete and well executed—just not meeting my standard for an ICLR publication.
> > > >
> > > > As a result, I am maintaining my original judgement: 4: marginally below the acceptance threshold. But would not mind if paper is accepted.

---

> ### Author Response · Authors · 2025-11-27
>
> Thank you again for your thoughtful response. Paralinguistic features have always been a key distinction between speech and text, and I fully appreciate your attention to this issue. With your permission, I would like to share some small experiments we conducted, which also help clarify our reasoning behind the final choice of paralinguistic types in this work.
>
> - **Speaker Identity**
>
>     We greatly appreciate your mention of multi-speaker scenarios, which indeed pose challenges for a speech system to respond based solely on a single speaker’s identity. However, our main concern is “**whether speaker identity can be inferred purely from text.**” Unless a speaker only utters a single line and then remains silent, it is often possible to deduce who spoke a particular utterance in recent turns using text alone.
>
>     To validate this, we randomly selected 200 sessions (1,809 turns) of multi-party conversations from MELD [1] and used Gemini 2.5 Pro to re-annotate speaker IDs based solely on textual content. The accuracy reached **73.2%**, confirming our concern that speaker identity can be largely inferred from text. Consequently, we believe that speaker identity is not well-suited as a metric for long-context audio benchmarks.
>
> - **Emotion**
>
>     Emotional information can be expressed through multiple modalities, including text and audio. While we have emphasized that emotion is an important paralinguistic feature in speech, it is important to note that text is often the most direct way to convey emotion. Many studies [1] on emotion recognition have shown that textual modality plays a critical role in accurate emotion identification, sometimes even outperforming audio.
>
>     To support this, we randomly selected 400 turns from MELD [1] for emotion recognition with Gemini 2.5 Pro. The results demonstrate that text (61.5%) can more effectively convey the emotions than audio (52.25%). Moreover, because textual input does not suffer from the common “cognitive degradation” issue seen in MLLMs, recognition performance can even be better.
>
>     |  | Audio (correct) | Audio (incorrect) | Total |
>     | --- | --- | --- | --- |
>     | Text (correct) | 160 (40%) | 86 (21.5%) | 246 (61.5%) |
>     | Text (incorrect) | 49 (12.25%) | 105 (26.25%) | 151 (38.5%) |
>     | Total | 209 (52.25%) | 191 (47.75%) | 400 (100%) |
>
>     An alternative approach, like the RAVDESS dataset [2], is to have the same semantic content read in different emotional styles. However, we are concerned that such forced simulation may fail to reflect real-world scenarios, as LALMs might also rely on semantic content when analyzing emotions. Thus, we think emotion less suitable for a long-context audio benchmark.
>
>
> ---
>
> In summary, we carefully considered different paralinguistic features and ultimately chose audio events to evaluate models’ acoustic understanding. We truly appreciate your suggestions regarding paralinguistic attributes; incorporating them into a benchmark has always been a goal we are interested in exploring further. We would be grateful for any feedback or suggestions you may have based on the experiments and considerations we shared.
>
> Once again, thank you for your valuable advice. Your feedback has greatly improved the rigor of our work. We look forward to your further comments and insights, and **if you agree with our reasoning regarding speaker identity and emotion, we would greatly appreciate your positive evaluation**.
>
> [1] MELD: A Multimodal Multi-Party Dataset for Emotion Recognition in Conversations. ACL 2018
>
> [2] The Ryerson Audio-Visual Database of Emotional Speech and Song (RAVDESS)

---

### Meta-Review · Area_Chair_5TN1 · 2026-01-05

**Summary:**

1. [cPLr] The proposed benchmark fails to evaluate models for their ability to process paralinguistic information such as speaker identity and emotion, which is not an appropriate choice for a speech benchmark.
2. [cPLr] The paper results do not make a strong case for the hybrid modeling approach.
3. [LqdZ, EdSM, inoe] Limited numbers of LALMs, as only 3 open-source models are provided.
4. [LqdZ] Why, in Table 1, Length Distribution for Conversational Dialogue presents 0-2min and 2-4min, but in Table 2, numbers for 4-8min and 8min+ are also provided?
5. [EdSM] Section 4.2 dedicates significant space to a relatively incremental position encoding method, not within the scope of a benchmark paper.
6. [EdSM] Relying solely on GPT-4o for accuracy assessment without human evaluation baseline or inter-rater agreement is problematic.
7. [EdSM] Table 2 is dense with too many columns making trends hard to identify.
8. [EdSM] While latency figures are provided, the paper lacks a discussion on memory requirements.
9. [inoe] The benchmark focuses mainly on QA tasks; adding generative or reasoning evaluations would strengthen coverage of long-context language understanding.
10. [inoe] MoPE’s algorithmic details lack ablation on parameter sensitivity; reporting computational cost variations would clarify its efficiency and reproducibility.

**Reviewer Concerns:**

1. This concern is not addressed in the rebuttal and revision. The authors defend the design of the benchmark, which excludes speaker identity and emotion, by arguing that both speaker identity and emotion can readily be inferred from text. In the discussion with Reviewer cPLr they provide some experimental results to support the claim, showing a speaker annotation accuracy of 73.2% from text and more accurate emotion recognition based on text than on audio by Gemini-2.5-Pro, evaluating on multiparty conversations from MELD.

I concur with Reviewer cPLr's opinion that omitting speaker identity and emotion substantially limits the value of the proposed benchmark. While awareness of background acoustic events and conditions can be expected to be useful in applications of spoken language models, the ability to track speaker identity and emotional state are much more valuable, and the text-based performance results cited by the authors fall below the QA accuracy they show in Table 2 in a number of cases. This is, in my mind, the most serious shortcoming of the paper.

2. This concern is fully addressed in the rebuttal and revision: the authors add results on the Beyond-Semantic Dialogues component illustrating the effectiveness of the hybrid context.

3. This concern is fully addressed in the rebuttal and revision: the authors added results for additional models.

4. This concern is fully addressed in the rebuttal and revision: it was simply a typographical error.

5. This concern is addressed in the rebuttal: the authors justify MoPE as an enabling technology for hybrid context and argue that they do not present it as a main contribution of the paper.

6. This concern is fully addressed in the rebuttal and revision: the authors add experimental results showing a close match between GPT-4o scoring and human evaluation.

7. This concern is fully addressed in the rebuttal and revision by a reworking of Table 2 with relegation of the detailed breakdown of results to an appendix.

8. This concern is fully addressed in the rebuttal and revision: the authors add measurements of GPU memory usage.

9. This concern is partially addressed in the rebuttal: the authors point out that two components of the benchmark focus on generative QA and they argue that reasoning is a topic for subsequent work.

10. This concern is addressed in the rebuttal: the authors point out that the only parameter is k, the number of preserved audio turns, and that Table 3 provides results showing how performance changes as k is varied.

**Reviewer Scores:**

**cPLr** - I do not believe that this reviewer would have changed their score because they were clearly dissatisfied with the authors' response on speaker identity and emotion being derivable from text.

**LqdZ** - I do not believe that this reviewer would have increased their already high score for this paper.

**EdSM** - I think it is possible that this reviewer would have increased their score, since their concerns were mostly addressed by the authors.

**inoe** - I think this reviewer would have retained their score, because I do not believe they would have been completely satisfied with the answer regarding generative and reasoning tasks. Specifically, I believe the reviewer had tasks other than QA in mind when they suggested looking at generative tasks.

---

### Decision · Program_Chairs · 2026-01-26

Reject